# Low Preconception Complement Levels Are Associated with Adverse Pregnancy Outcomes in a Multicenter Study of 260 Pregnancies in 197 Women with Antiphospholipid Syndrome or Carriers of Antiphospholipid Antibodies

**DOI:** 10.3390/biomedicines9060671

**Published:** 2021-06-11

**Authors:** Cecilia Nalli, Daniele Lini, Laura Andreoli, Francesca Crisafulli, Micaela Fredi, Maria Grazia Lazzaroni, Viktoria Bitsadze, Antonia Calligaro, Valentina Canti, Roberto Caporali, Francesco Carubbi, Cecilia Beatrice Chighizola, Paola Conigliaro, Fabrizio Conti, Caterina De Carolis, Teresa Del Ross, Maria Favaro, Maria Gerosa, Annamaria Iuliano, Jamilya Khizroeva, Alexander Makatsariya, Pier Luigi Meroni, Marta Mosca, Melissa Padovan, Roberto Perricone, Patrizia Rovere-Querini, Gian Domenico Sebastiani, Chiara Tani, Marta Tonello, Simona Truglia, Dina Zucchi, Franco Franceschini, Angela Tincani

**Affiliations:** 1Rheumatology and Clinical Immunology Unit, ASST Spedali Civili and University of Brescia, 25123 Brescia, Italy; danielelini@gmail.com (D.L.); laura.andreoli@unibs.it (L.A.); crisafulli.francesca10@gmail.com (F.C.); fredi.micaela@gmail.com (M.F.); mariagrazialazzaroni@gmail.com (M.G.L.); francofranceschini1@gmail.com (F.F.); angela.tincani@unibs.it (A.T.); 2Department of Clinical and Experimental Sciences, University of Brescia, 25123 Brescia, Italy; 3Department of Obstetrics and Gynecology, I.M. Sechenov First Moscow State Medical University of the Ministry of Health of the Russian Federation (Sechenov University), Moscow 119992, Russia; vikabits@mail.ru (V.B.); totu1@yandex.ru (J.K.); gemostasis@mail.ru (A.M.); 4Rheumatology Unit, Department of Medicine, University of Padua, 35100 Padua, Italy; antonia.calligaro@gmail.com (A.C.); teresa.delross@aopd.veneto.it (T.D.R.); mariafavaro@gmail.com (M.F.); 5Division of Immunology, Transplantation and Infectious Diseases, IRCCS San Raffaele Scientific Institute, 20121 Milan, Italy; canti.valentina@hsr.it (V.C.); rovere.patrizia@hsr.it (P.R.-Q.); 6Division of Clinical Rheumatology, ASST Gaetano Pini-CTO Institute, 20122 Milan, Italy; roberto.caporali@unimi.it (R.C.); maria.gerosa@unimi.it (M.G.); 7Department of Clinical Sciences and Community Health, Università degli Studi di Milano, 20122 Milano, Italy; cecilia.chighizola@unimi.it; 8Rheumatology Unit, Department of Biotechnological and Applied Clinical Science, School of Medicine, University of L’Aquila, 67100 L’Aquila, Italy; francescocarubbi@libero.it; 9Research Center for Adult and Pediatric Rheumatic Diseases, Pediatric Rheumatology Unit, ASST G. Pini & CTO, 20122 Milan, Italy; 10Rheumatology, Allergology and Clinical Immunology, Department of “Medicina dei Sistemi”, University of Rome Tor Vergata, 00133 Rome, Italy; paola.conigliaro@uniroma2.it (P.C.); roberto.perricone@uniroma2.it (R.P.); 11Lupus Clinic, Reumatologia, Dipartimento di Scienze Cliniche Internistiche, Anestesiologiche e Cardiovascolari, Sapienza Università di Roma, 00185 Rome, Italy; fabrizio.conti@uniroma1.it (F.C.); dott.trugliasimona@gmail.com (S.T.); 12Polymedical Center for Prevention of Recurrent Spontaneous Abortion, 00168 Rome, Italy; cdecarolismd@gmail.com; 13Rheumatology Unit, Azienda Ospedaliera San Camillo-Forlanini, 00152 Rome, Italy; annamariaiuliano@hotmail.it (A.I.); GSebastiani@scamilloforlanini.rm.it (G.D.S.); 14Istituto Auxologico Italiano, 20122 Milan, Italy; pierluigi.meroni@unimi.it; 15Department of Clinical and Experimental Medicine, University of Pisa, 56126 Pisa, Italy; marta.mosca@med.unipi.it (M.M.); chiaratani78@gmail.com (C.T.); dinazucchi@hotmail.it (D.Z.); 16Rheumatology Unit, Department of Medical Sciences, University of Ferrara and Azienda Ospedaliero-Universitaria S. Anna, 44124 Ferrara, Italy; melissa.padovan@unife.it; 17BSC, Rheumatology Unit, Department of Medicine, University of Padua, 35100 Padua, Italy; marta.tonello@unipd.it

**Keywords:** pregnancy, complement, antiphospholipid antibodies, antiphospholipid syndrome, gestational outcome

## Abstract

Antiphospholipid antibodies (aPL) can induce fetal loss in experimental animal models. Human studies did find hypocomplementemia associated with pregnancy complications in patients with antiphospholipid syndrome (APS), but these results are not unanimously confirmed. To investigate if the detection of low C3/C4 could be considered a risk factor for adverse pregnancy outcomes (APO) in APS and aPL carriers’ pregnancies we performed a multicenter study including 503 pregnancies from 11 Italian and 1 Russian centers. Data in women with APS and asymptomatic carriers with persistently positive aPL and preconception complement levels were available for 260 pregnancies. In pregnancies with low preconception C3/C4, a significantly higher prevalence of pregnancy losses was observed (*p* = 0.008). A subgroup analysis focusing on triple aPL-positive patients found that preconception low C3 and/or C4 levels were associated with an increased rate of pregnancy loss (*p* = 0.05). Our findings confirm that decreased complement levels before pregnancy are associated with increased risk of APO. This has been seen only in women with triple aPL positivity, indeed single or double positivity does not show this trend. Complement levels are cheap and easy to be measured therefore they could represent a useful aid to identify patients at increased risk of pregnancy loss.

## 1. Introduction

Antiphospholipid syndrome is a rare autoimmune disease characterized by thrombotic events involving the venous and arterial systems, including microcirculation, and/or pregnancy morbidities in the presence of confirmed positivity for aPL [1]. aPL are mainly directed against phospholipid-binding proteins and the main antigenic target for aPL is beta2glycoprotein I (B2GPI), a protein found on lipid layers in cellular membranes [2]. The complement system has attracted attention as a potential mediator of pathogenic mechanisms induced by aPL and its activation is regarded as a necessary event not only for thrombosis but also for obstetric complications [3,4]. The complement system comprises over 30 proteins that act in concert to protect the host against invading organisms. Its activation can be triggered via three different pathways: classical, alternative, and leptin. All the pathways converge on the C3 protein and cleave to generate fragments C3a and C3b. C3b attaches covalently to targets, followed by assembly of C5 convertase and the subsequent cleavage of C5 to C5a and C5b [5]. Since the mid-1990s, it has been investigated that activated complement fragments have the capacity to bind and activate inflammatory and endothelial cells in vivo, as well as to induce a prothrombotic phenotype. Several mechanisms have been proposed to account for aPL-induced activation of complement. In particular, anti-B2GPI antibodies-B2GPI complexes have been shown to activate the complement cascade.

In obstetric APS, evidence of complement role has been gained in animals treated with aPL fractions. Girardi et al. demonstrated that C5 deficiency or treatment of mice with anti-C5a monoclonal antibody protects against aPL-induced pregnancy loss and growth retardation [3]. Another group in the same year published that inhibition of the complement cascade in vivo using the C3 convertase inhibitor (Crry-Ig) prevents fetal loss and growth restriction [6]. Furthermore, mice deficient in C3, C4, C5, and factor B were not prone to develop aPL-induced fetal loss [7,8,9]. The progressive evidence of complement involvement in aPL-related pregnancy loss derived by animal models prompted several groups to investigate the significance of complement levels in human disease. B2GPI, the recognized main target of aPL, is widely represented on trophoblast and decidual surface [10]. Complement C3 and C4 serum levels were then assessed in several cohorts of pregnant patients with APS and/or aPL in order to relate complement consumption with APO. However, these studies have yielded inconsistent results, in fact, while some studies have come to find a correlation [11,12], other studies have not revealed a prognostic role for the complement in relation to pregnancy morbidity among aPL-positive women [13,14]. More recently, complement activation products were found to be increased during pregnancy in patients with aPL and APO by two different groups [15,16].

This multicenter retrospective study was conducted to further clarify the prognostic role of preconception serum C3 and C4 levels in a cohort of APS and/or aPL carrier pregnant women without any underlying autoimmune disease.

## 2. Materials and Methods

### 2.1. Study Cohort and Inclusion/Exclusion Criteria

Medical records of pregnant women with confirmed positivity for aPL antibodies attending twelve referral centers from January 2010 to December 2020 were retrospectively evaluated.

Exclusion criteria were the presence of an associated systemic autoimmune disease, diagnosed according to the international classification criteria, voluntary termination of pregnancy, fetal losses due to severe fetal malformations. Demographic and clinical data were collected in an anonymized ad hoc created database:-criteria and non-criteria manifestations of APS [1];-aPL profile and C3 and C4 levels at diagnosis and preconception (considered at least 6 months before pregnancy);-therapy during the three trimesters of pregnancy;-gestational outcome and maternal complications.

### 2.2. APO Definitions

For the purpose of this study, we considered as aPL-related APO: spontaneous abortions (<10 weeks of gestation), fetal loss (≥10 weeks of gestation), neonatal death (death of a formed fetus alive at birth in the first 28 days of life), preterm delivery before 37 weeks of gestation, preeclampsia, eclampsia, or HELLP syndrome (hemolysis, elevated liver enzymes, and low platelet). We excluded pregnancies with other recognized causes for adverse pregnancy outcomes (i.e., therapeutical abortion consequent to the finding of anatomical abnormalities).

### 2.3. Autoantibody Detection

aPL were tested by each participating center in a referral laboratory. Anticardiolipin antibodies (aCL) and antiB2GPI antibodies (aB2GPI) were detected by ELISA according to the current recommendations [17]. Lupus anticoagulant (LA) was detected by coagulation assay according to the guidelines of the International Society on Thrombosis and Hemostasis [18]. Complement C3 and C4 fractions were detected as in clinical practice.

### 2.4. Statistical Analysis

Categorical variables were reported as a proportion and/or percentage, whereas continuous variables as mean (±standard deviation) values. Fisher’s exact test or chi-squared test for categorical variables and Mann–Whitney test for continuous variables were applied as appropriate. All tests were performed using GraphPad Prism 9. Logistic regression was applied for multivariate analysis using Statview. *p*-values ≤ 0.05 were considered significant and odds ratio (OR) with 95% confidence interval (95% CI) was indicated.

## 3. Results

We collected data about 503 pregnancies in 383 patients. The patients were Caucasian (n = 342, 89.2%), African Americans (n = 21, 5.5%), Asian (n = 14, 3.7%), and Latin American (n = 6, 1.6%). In 333 patients (86.9%), a diagnosis of APS according to the classification criteria [1] had been formulated, while 50 (13.1%) were aPL carriers.

Most of the women (228, 68.5%) presented with obstetric morbidity only, while 105 patients (31.5%) had experienced thrombotic events, with or without pregnancy morbidity.

In this cohort, 260 singleton pregnancies in 197 patients with available preconception complement levels and gestational outcomes (52%), 76/143 aPL carriers (53%), and 184/360 (51%) APS pregnancies, were available. A total of 93 (36%) of all pregnancies had low levels of preconception C3 (51/93, 55%) or C4 (13/93, 14%) or both (29/93, 31%). A total of 167 (64%) pregnancies had normal complement levels.

### 3.1. Autoantibodies Profile

The results of the three aPL assays were available for all the patients. LA was detectable in 97 pregnancies (37.3%). aCL IgG were positive in 180 pregnancies (69.2%), and IgM in 77 (29.6%); anti-B2GPI IgG antibodies were positive in 110 pregnancies (42.3%) and IgM in 96 (36.9%). A triple aPL positivity was observed in 62 women (23.8%) while double in 55 (21.2%) and single in 143 (55%). Antinuclear antibodies (ANA) were persistently positive in 71 patients (27%), anti-dsDNA in 8 (8.6%), anti-extractable nuclear antigens (anti-ENA) in 9 (3.5%), and anti-thyroperoxidase antibody (anti-TPO) and/or anti-thyroglobulin antibody (anti-TG) in 24 (9.2%).

### 3.2. Pregnancy Outcome

Most pregnancies (224, 86.2%) culminated with a live birth, at a mean gestational age of 37.6 ± 3.4 weeks (range 25.6–41.5). As detailed in Table 1, pregnancy loss occurred in 36 gestations.

During follow-up APO were described in 94 APS and 21 aPL carriers’ pregnancies (51% and 29% respectively). Patients with APO showed significant lower complement levels than women with uncomplicated pregnancies (Figure 1).

Considering the 93 patients (71 APS and 22 aPL carriers) with low preconception C3 and/or C4 and comparing them to patients with normal complement level, a significantly higher prevalence of pregnancy losses was observed (*p* = 0.008) as well as a higher prevalence of preterm live birth from the 37th week of gestation and earlier (Table 2).

Furthermore, we performed subgroup analysis, considering separately patients with triple aPL positivity. It is worthwhile to underline that in these 62 patients decreased C3/C4 were extremely common (77.4%). In this group of high-risk patients, preconception low C3 and/or C4 levels were found to be associated with an increased rate of pregnancy loss (*p* = 0.05). On the other hand, among women with single or double aPL positivity, APO was not related to preconception complement levels (Table 2). Maternal complications (preeclampsia n = 14, deep vein thrombosis n = 3, and thrombocytopenia n = 6) were not statistically related with low preconception levels of C3 and/or C4.

In multivariate analysis, the only feature associated with complicated pregnancies was the preconception triple positivity for aPL, both in APS and aPL carrier group (*p* = 0.02, OR 2.421, CI 95% 1.112–5.273 and *p* = 0.03, OR 5.823, CI 95% 1.120–30.277, respectively). C3 and C4 preconception levels did not show any correlation, as well as maternal diagnosis.

### 3.3. Treatment

Most patients with APS were treated with low-dose-aspirin (LDA, n = 161; 87.5%) and/or low-molecular-weight heparin (LMWH; n = 158; 85.8%) during pregnancy. Immunomodulatory or immunosuppressive therapy was recorded in 40 pregnancies, with hydroxychloroquine (HCQ) administrated in 38 cases (20.6%) and low-dose corticosteroids (CS) in 8 (4.3%).

Pregnancies in aPL carriers were treated with LDA in 64 cases (84.2%) and LDA and/or LMWH in 28 (26.8%). In these patients, HCQ was administrated in 4 (5.2%) pregnancies and CS in 5 (6.5%).

Combination therapy with LDA and LMWH was more frequent in patients with triple aPL positivity compared to single/double positivity (82.2% vs. 59.7%, respectively, *p* = 0.001). Moreover, combination therapy was used more frequently in patients satisfying the criteria for primary APS compared to aPL carriers (53.6% vs. 33.8%, *p* = 0.005). In multivariate analysis patients with complicated pregnancies were more frequently treated with combination therapy, LDA+LMWH (*p* = 0.005, OR 2.200, CI 95% 1.273–3.800); however, it did not relate with low preconception C3/C4 levels.

Lastly, we found in patients with triple aPL positivity (with and without APS) and complement consumption that the administration of HCQ on top of combination therapy during pregnancy was significantly related with a better gestational outcome compared to patients that had received only LDA+LMWH (70% vs. 23% did not present any APO, *p* = 0.018). This observation could not be confirmed in patients with single or double aPL positivity.

## 4. Discussion

This multicenter study allowed us to identify preconception decreased C3 and C4 levels as a predictive marker for the occurrence of APO in aPL-positive patients with or without clinical manifestations. In a large sample of 260 pregnancies, a decrease in preconception levels of C3/C4 levels were found in 36% of the patients. Overall, women with APO showed significant lower preconception complement levels than those with successful pregnancies, without any difference between APS and aPL carriers. As shown in univariate analysis, low preconception complement levels, mainly C3, resulted as significant risk factor for prematurity and pregnancy loss. This finding is in agreement with previous studies showing an association between low C3/C4 levels and APO [14,16,19]. Partially consistent data were raised by Deguchi and coworkers, who observed that only hypertensive pregnancy complication of APS but not fetal loss are related to decreased C3/C4 levels [14]. Other authors described lower complement levels in APS pregnant women compared to the obstetrical general population but without any relationship with pregnancy loss [13].

Unfortunately, findings emerging in different studies are not always easily comparable: complement determinations were performed in different weeks of gestation and it is well known that pregnancy itself exerts an important influence on the complement components synthesis [20,21,22,23]. Heterogeneity refers even to clinical criteria for study inclusion, with some studies enrolling exclusively women with primary APS and others also women with systemic autoimmune rheumatic conditions. To note, in our study patients with concomitant autoimmune diseases, especially systemic lupus erythematosus, were excluded to avoid bias. ANA positivity was found in 27% of cases, but none of the patients had any additional sign or symptom of SLE or other systemic autoimmune diseases. Autoimmune thyroid disease was found in 9.2%, partially accounting for the high ANA positivity rate.

If animal models have clearly shown that activation of complement is required to produce aPL-mediated pregnancy loss and inflammation and complement deposition at trophoblast and placental level was described [10], histopathological data from human disease are not consistent. In fact, human placenta analysis does not unanimously show complement deposition, and inflammation (chorioamnionitisis or villitis) does not seem always associated with aPL-mediated pregnancy loss [24,25].

This work also addressed the differential prognostic role of complement levels in patients stratified upon the aPL profile (triple versus single/double aPL positivity). The profile of triple aPL positivity is well known to identify patients at high risk for pregnancy loss [26]. Consequently, in this setting the combination therapy during pregnancy is strongly supported not only in patients with obstetrical APS only (without previous thrombosis) but also in aPL carriers, at least when multiple risk factors are identified [11,27,28]. In our study, triple aPL positivity was recorded in 23.8% of the whole cohort, most of the triple aPL-positive patients displayed low C3/C4 levels (77%). Despite the bias related to a better treatment approach and despite the relatively low number of triple aPL-positive pregnancies, low preconception levels of C3 and/or C4 significantly relate to pregnancy loss (*p* = 0.05). In the group of patients with single/double positivity, the pregnancy outcome was not related to the complement levels before pregnancy.

We also could confirm, in multivariate analysis, the association between APO and triple aPL positivity. Such association held significance even if in most pregnancies (81%) combination therapy (LDA and LMWH) was instituted. This finding suggests that pregnancies in patients with triple aPL positivity, independently from maternal diagnosis and from conventional treatment, seems to be more often characterized by complications. Recently, women with high-risk aPL profiles were found to have a probability of pregnancy morbidity as high as 52% despite conventional treatment (EUREKA). Not surprisingly, in this particular high-risk subgroup, the addition of immunomodulatory therapy has been suggested in literature, ranging from HCQ [29,30] as well as low-dose corticosteroids or intravenous immunoglobulins [31,32]. In our cohort, the therapeutic choices were formulated by the attending physicians, based on the clinical and laboratory profile as well as the previous obstetrical history of each patient. In particular, previous pregnancy failures despite combination therapy and the presence of the so-called “non criteria manifestations”, which have been associated with possible poor gestational outcome [28], can support adding HCQ on top of conventional therapy. Of note, we observed that in triple aPL-positive patients with complement consumption, the addition of HCQ to the combination therapy is linked to a significantly reduced rate of APO. This positive effect on pregnancy complications was not observed in patients with normal complement levels. Positive effects of HCQ during pregnancy in APS patients were already described by other authors and suggested by EULAR recommendations that propose this additional treatment option [29,33].

We reported a low rate of maternal complications during pregnancy and/or puerperium. In particular, we reported three mothers with deep vein thrombosis. One patient with vascular APS suffered a deep vein thrombosis during the second trimester, while she was on LMWH at therapeutic dosage. Two women, one vascular and one obstetric APS, experienced venous thrombosis during puerperium in accordance with the well-known risk of postpartum thrombosis in the general obstetric population. In both cases, the patients were in prophylactic therapy with LMWH.

We also observed 14 pregnancies (5%) complicated by PE/HELLP syndrome, 6 of them were patients with a triple positivity for aPL (4 APS).

In the group of pregnancies with maternal complications, we did not find any statistical correlation with low preconception C3/C4 levels in contrast with what has been reported by other authors [34].

This study has several limitations: the retrospective design, which could have led to possible completeness and registration bias; the lack of a centralized laboratory, which could be considered not so relevant since only routine assays were included in the study; the multicenter nature, a possible source of heterogeneity.

However, the study also has several strengths: the inclusion of a large number of pregnancies in patients with aPL/APS, regularly followed throughout their gestational period; the useful application of simple and cheap laboratory tests, such as C3 and C4 levels that are routinely and widely performed; the inclusion of patients from twelve different centers, which testifies to the solid nature of the obtained results.

In conclusion, this study shows that low levels of C3 and C4 in aPL/APS patients are linked to a worse pregnancy outcome, even in patients with triple antibody positivity, which already carries a bad prognosis. Therefore, C3 and C4 complement assay, which are inexpensive and routinely available, could provide a valid tool to better stratify the risk of pregnancy morbidity in women carrying aPL. Given the high rate of unresponsiveness to treatment we still observe among aPL-positive women embarking on a pregnancy, it would be valuable to early identify women that could possibly benefit from a closer monitoring and a more aggressive therapeutic approach, such as the addition of HCQ to conventional treatment.

## Figures and Tables

**Figure 1 biomedicines-09-00671-f001:**
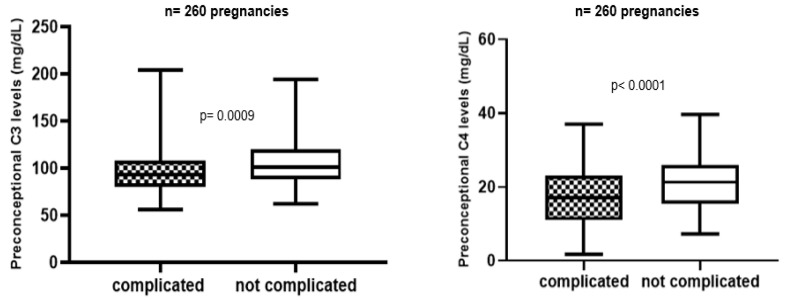
Preconception complement level in complicated vs. not complicated pregnancies.

**Table 1 biomedicines-09-00671-t001:** Outcome and APO of the study cohort.

Gestational Outcome and Maternal Complications	260 Pregnancies (N, %)
Spontaneous abortion	27 (10.4%)
Fetal death	7 (2.7%)
Live births	224 (86.2%)
Neonatal death	2 (0.8%)
Preterm deliveries < 37 weeks	77 (29.6%)
SGA ^a^	3 (1.2)
Intrauterine growth restriction	5 (1.9)
Preeclampsia/HELLP syndrome ^b^	14 (3.9)
Thrombocytopenia	6 (2.3)
DVT ^c^	3 (1.1)
Gestational diabetes	6 (2.3)
PROM ^d^	9 (3.5)
Hemolytic anemia	1 (0.4)

^a^ SGA: small for gestational age was defined as a birth weight in the <10th percentile for gestational age. ^b^ HELLP syndrome: concomitant presence of severe thrombocytopenia (platelets ≤ 50,000/μL), evidence of hepatic dysfunction (liver enzymes ≥70 IU/l), and evidence suggestive of hemolysis (total serum lactate dehydrogenase ≥600 IU/l). ^c^ DVT: deep vein thrombosis. ^d^ PROM: preterm premature rupture of membranes was defined as rupture of the membranes before 37 weeks of gestation.

**Table 2 biomedicines-09-00671-t002:** Relationship between gestational outcome, maternal pregnancy complications and preconception complement levels.

	All Pregnancies (n = 260)				Triple aPL Positivity (n = 62)				Single or Double aPL Positivity (n = 198)			
Gestational Outcome	Low C3/C4 (n = 93)	Normal C3/C4 (n = 167)	*p*	OR (CI 95%)	Low C3/C4 (n = 48)	Normal C3/C4 (n = 14)	*p*	OR (CI 95%)	Low C3/C4 (n = 45)	Normal C3/C4 (n = 153)	*p*	OR (CI 95%)
Term live birth (≥37 w)	39 (42%)	121 (72%)	<0.0001	0.367 (0.205–0.655)	13 (27%)	7 (50%)	ns	-	26 (58%)	114 (75%)	ns	-
Preterm live birth (<37 w)	34 (37%)	30 (18%)	<0.0001	2.390 (1.337–4.274)	23 (48%)	7 (50%)	ns	-	11 (24%)	23 (15%)	ns	-
Pregnancy losses (abortion, fetal death, and neonatal death)	20 (21%)	16 (10%)	0.008	2.586 (1.266–5.282)	12 (25%)	0 (0%)	0.05	-	8 (18%)	16 (10%)	ns	-

## Data Availability

Data will be available upon reasonable request.

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
