# Peer review of "Low Preconception Complement Levels Are Associated with Adverse Pregnancy Outcomes in a Multicenter Study of 260 Pregnancies in 197 Women with Antiphospholipid Syndrome or Carriers of Antiphospholipid Antibodies"

_biomedicines, 2021, doi:10.3390/biomedicines9060671_

Round 1

Reviewer 1 Report

In their study Nalli et al, performed a multicenter study that included 503 pregnancies from Italian and Russian Centers. The authors show that the low levels of C3 and C4 complement components in antiphospholipid antibodies/ antiphospholipid syndrome. The general conclusion that testing the levels of C3 and C4 complement components can be a good prognostic and diagnostic tool, in particular, less costly.

Complement system factors and components have a role in platelet function. APS syndrome is linked in many diseases to platelet activation and thrombosis.

I think the manuscript is well written, in the discussion the authors discuss venous thrombosis, and therapies using a low molecular weight of heparin. However, there is no so much about platelets in the manuscript. In APS, they have an important role. Also please discuss the different cellular sources of C3 and C4. All these aspects can be discussed in the manuscript.

Author Response

Dear reviewer thank you very much for your questions.

In our cohort we described 6 patients with mild thrombocytopenia (2.3%) and all these pregnancies had a favorable pregnancy outcome without any other complications (maternal or fetal) reported. Almost all patients were treated during pregnancy with LDA as for recommendations.  

C3 and C4 complement products are synthetized mainly by the liver (even if a low proportion of the complement circulating proteins is produce in other tissue like the kidneys).  In this article we focused on clinical aspects and we have chosen to do not discuss the synthesis of the various component of the complement cascade; the same choice has been done for aPL. 

Reviewer 2 Report

Minor revisions for the authors:

  • Misprint in abstract: “This it has been seen only in women with triple aPL positivity,…”
  • In table 2 you write: “IC 95%” You mean “CI 95%”.
  • In the Discussion you write: “ANA positivity was found in 27% of cases, but all patients had any additional sign or symptom of SLE or other systemic autoimmune diseases.” Do you mean: “…none of the patients had any sign of SLE…”

Major revisions for the authors:

  • You write in the abstract and in the results: “A subgroup analysis focusing on triple aPL positive patients found out that preconception low C3 and/or C4 levels was associated with an increased rate of pregnancy loss (p=0.05)” and “In this group of high risk patients, preconception low C3and/or C4 levels were found associated with increased rate of pregnancy loss (p=0.05)” but also “P-values < 0.05 were considered significant” in Statistical analysis. The conclusion of an association is not accurate since the p value is not below 0.05.
  • You mention the patients were of different ethnicities. Did you find any correlation there with the results? Does ethnicity have an impact on the outcome? Did you adjust the results for the ethnicity or other baseline values?
  • Many of the patients had more than one pregnancy in the study (260 pregnancies in 197 females). Was there a difference between the pregnancies in the same female?

Round 2

Reviewer 2 Report

Dear Editor,

I believe the manuscript is very interesting, comprehensive, well written and very much publishable.

Best regards,

Shahrzad